# Marine Geophysical Data and Its Application to Assessment of Crustal Structure along the Northern Egyptian Passive Continental Margin

Hamdy A. M. Aboulela [1,2]

1   Department of Marine Geology, Faculty of Marine Sciences, King Abdulaziz University, Jeddah 21589, Saudi Arabia; aboulel1_004@yahoo.co.uk
2   Marine Science Department, Faculty of Science, Suez Canal University, Ismailia 8366004, Egypt

**Abstract:** The Egyptian passive continental margin is considered a remarkable geologic setting, in addition to being an occupation and manufacturing locality in north Egypt. This work used accessible potential field data, such as marine gravity data, to provide a wider vision of the potential field of the area under investigation. The results of the two-dimensional (2D) gravity modelling revealed a good agreement between the modelled gravity and observed gravity fields, including known regional structures found in the investigated area. The findings revealed that crustal modelling was affected by the tectonic structure and the huge thickness of sedimentary layers, which act as barriers to the crystal crust. The results revealed that the crustal thickness and density are spread among the deposited layer and the inferior mantle in the Moho range. Furthermore, it was found that the basement extent lies nearly 6–9 km lower in the northern Egyptian coastline to approximately 13 km under the Herodotus abyssal plain. Moreover, it was shown that the thickness of the sedimentary layers deposit increases near the East Mediterranean Ridge.

**Keywords:** Continental margin; free-air gravity; regional and residual; crustal structure; two-dimensional modelling

## 1. Introduction

The area under investigation in this study has great significance as an economic site for Egypt's oil sector; thus, it is important to be able to distinguish offshore sedimentary basins from adjacent surface marine geological structures in this area (Figure 1a). In addition, this area is characterised by significant geological settings associated with tectonic activity evolutions [1–13]. The Egyptian passive continental border is a unique area distinguished by an enormous sedimentary categorization dating from the Pliocene to the present. The study area involved several major microstructural domains, such as the East Mediterranean ridge [3,9,14]. Previous studies have shown that the area under investigation appears to be characterised by changeability in the crustal construction due to the subduction of the sea bottom, and the appearance of numerous bathymetric anomalies in the ditches disrupts the subduction process [6,9]. Despite the conclusions provided in these studies, there are still uncertainties about the underlying crustal structure; meanwhile, the character of the dissimilar tectonic units is still up for debate. Moreover, there is only limited information relating to tectonic and geodynamical processes along a complex tetanisation region such as crustal structure, the transition between oceanic-continental crust, and the integrated models of the gravity field observed in the area investigated. This is not due to data deficiency, but rather is due to difficulties in data interpretation. Currently, geophysical studies with model techniques can be used to ascertain the crustal structure, and the source of the anomalies can be indirectly traced as a structural setting. In this work, a geophysical interpretation was carried out on the available marine gravity, seismic, bathymetric, and topographic data. Moreover, the integrated marine gravity field, seismic

data, and ancillary data, were examined using two-dimensional (2D) gravity modelling techniques. The aim of this work was to achieve a broad vision of the marine gravity field and bathymetric and topographic features of the Egyptian passive continental margin. To recognize the 2D gravity modelling data and define the crust, the level crustal structure, and its thickness, moreover the dissimilarity in the depth to the basement under the study area, were identified.

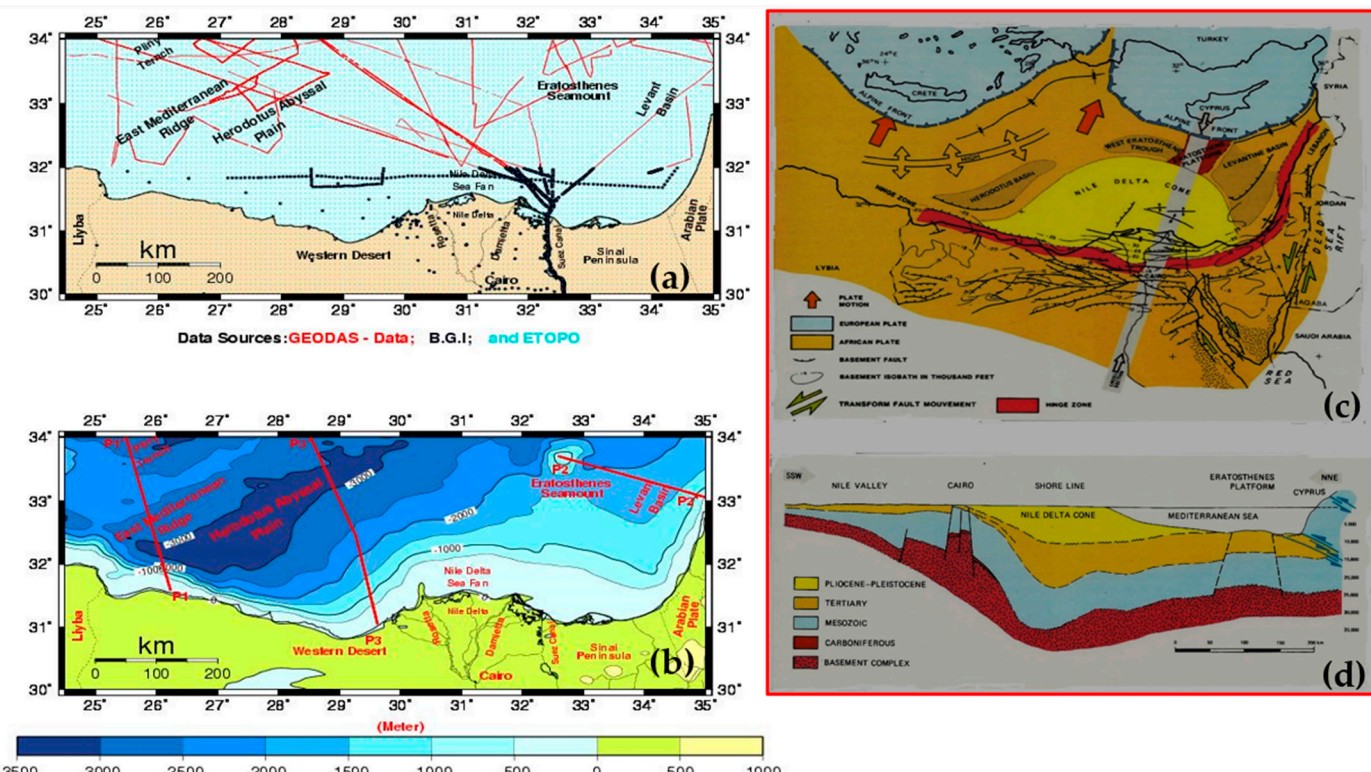

**Figure 1.** (**a**) Location map of the area under investigation. The heavy line mark indicates the site of collection for the data sources obtained from [15–17]. (**b**) Bathymetric and topographic feature map of the area investigated. The thick red line represents the site of the three seismic profiles (P1–P1″, P2–P2″ and P3–P3″). (**c**,**d**) Sketch showing regional tectonic setting in the study area and its environs; and the lower diagram shows a schematic cross-section along the line indicated in the upper part respectively modified from [18].

## 2. Regional Geological Setting

The regional geological setting of the area under study and its surroundings can be briefly described as follows.

The overall regional geological background of the study area is organized by a dynamic rift in the Red Sea and the Eurasia–Africa junction in the Mediterranean Sea. The plate boundary margins in the north section of Egypt include the Levant transform fault, the African-Eurasian plate and the Red Sea plate margins [8,18–22].

The study area is located to the south of the crumpled arc establishing the East Mediterranean Ridge and the Herodotus Basin, wherever the sea bottom is employed by the Nile Delta cone (Figure 1a,b). The major bathymetric and topographical landscapes are moderately widespread and display an inconsistency in size in the east–west direction of the coastline district together with the sea bottom. In addition, the study area closest to the southeastern Mediterranean Sea is characterized by a slight continental bookshelf spreading along coastline and shelf brink at approximately 15–20 km (Figure 1b).

The Nile Delta cone is a joint district that contains some southward graben drops. These graben drops are malformed and are then limited by east–west dipping faults positioned northwards (Figure 1c,d). The study area and its surroundings were formed

during the tectonic breakdown of the Pangea through the mid-Permian to the mid-Jurassic periods [5,20,23–25]. The Levantine basin was in a continental margin situation through the mid-Jurassic to upper Cretaceous. This stage was characterised by normal faulting, sub-parallel to the contemporary day southeast Mediterranean coastline, and Levantine basin. In late Cretaceous to Paleogene times, the region experienced compression as a consequence of the merging of the Eurasian-African plates. This led to the transposition of the previously formed NE trending grabens and to oblique slip faulting due to distinct plate motion. The major significant elements in the study area include the Eratosthenes Seamount, Levantine basin, Cyprus/Larnaca thrust zone and coastline boundary faults lateral to the eastern boundary of the sink. The East Mediterranean Ridge, Nile Delta cone and Levant Basin are the main morphostructural provinces.

The Mediterranean Ridge is extended through various prisms within southern Europe and Africa containing deposits derived from the subduction plate. Currently, a stress system at 30° northwest is disrupting the study area with extreme compressional stress, which includes normal faults trending almost through this trend in addition to rebooted reverse faults in the vertical trend. Generally, the type of the crust beneath this area is an outcome of the dissimilar variability of the crustal type, as reported in [22]; wherever the Levant basin remains, the oceanic crust related to its seismic revisions will create detects in the continental or semi-continental crust, as stated in [26–28]. The depth of the deposits on the northern Egyptian passive continental margin may be as deep as 9 km, as reported in [29].

## 3. Data Processing Methods and Its Result

To achieve the aims of this study, several accessible datasets, including marine gravity data, seismic data, bathymetric and topographic data, magnetic data, and other ancillary data, were sensibly revised and reviewed. Such an assessment will improve our understanding of the situation in the area under investigation. Several profiles of marine potential field data in this study were obtained from a databank, such as the GEODAS and BGI databases [15]. These data were integrated with data from the Egyptian General Petroleum Company [30], which were issued as a combination of Bouguer gravity maps of Egypt at a 1:500,000 scale obtained by collecting all accessible gravity data (Figure 2a). An elementary free-air gravity map was created to provide explanation of the investigated area (Figure 2b). To achieve morphotectonic relief, some profiles stayed as developed by the BGI. These data were collected from the available ETOPO database [16] to create the major geographies of the bathymetric and topographic map shown in Figure 2a. In addition, available magnetic data were acquired from the GEODAS database. [17]. A total magnetic intensity map (Figure 2c) was created to confirm a qualitative interpretation of the investigated area. The locations of the datasets are shown in different colours in Figure 1a. Moreover, to improve and evaluate the 2D model of marine gravity data, the results from deep seismic sounding experiments were acquired along with three seismic profiles (P1–P1″, P2–P2″, and P3–P3″) from the study area. The Geological Survey of Cyprus approved these investigations to determine the structure of the deep sedimentation throughout the study area. The three seismic regional construction outcome profiles were recycled to provide layer geometry and thickness. They allowed for the preliminary evaluations of the strata densities laterally; the three seismic profiles intersect the foremost geological feature of the area under investigation and were used to assess a tectonic model in support of this district. The locations of the three seismic profiles are shown by thick red lines (Figure 1b).

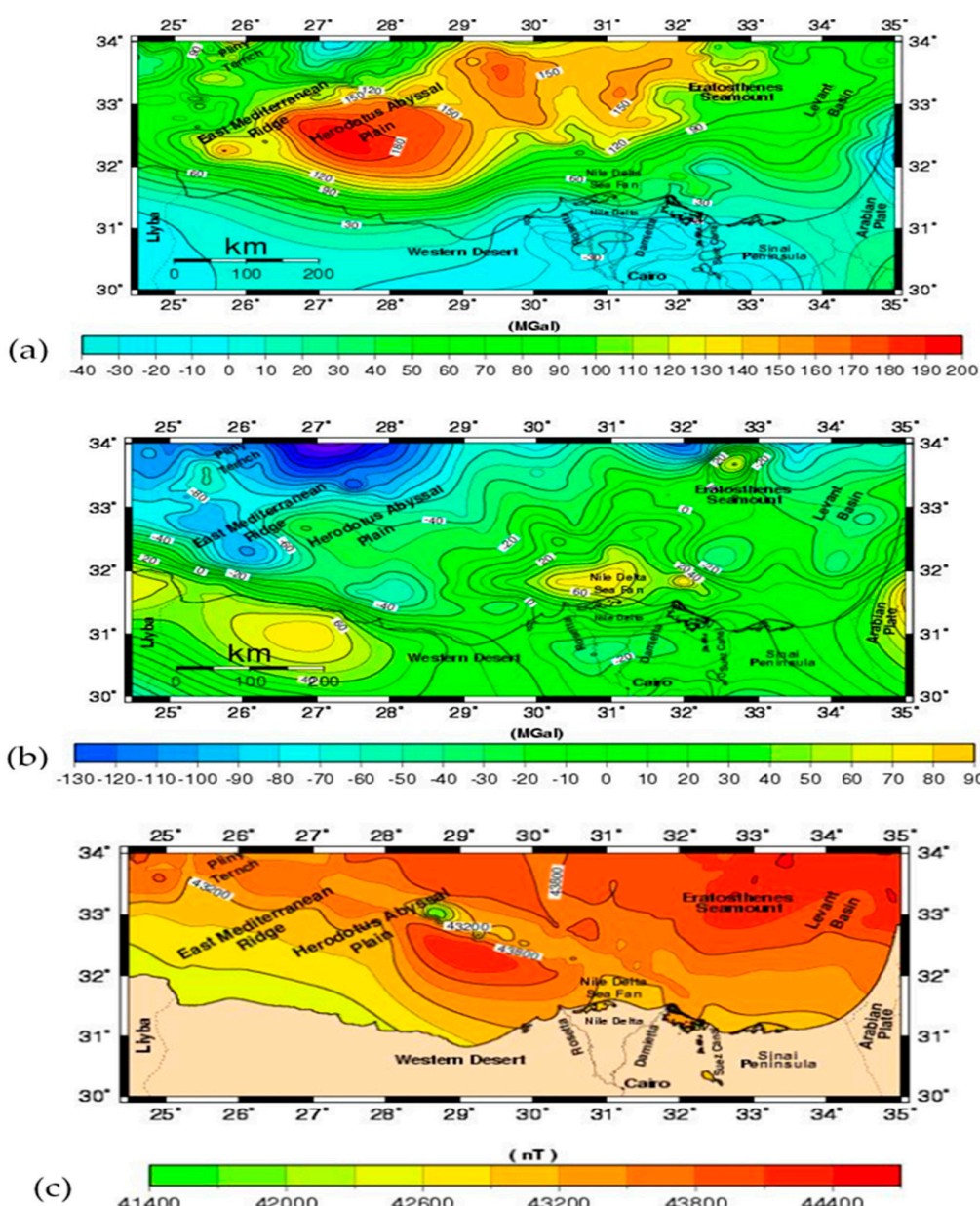

**Figure 2.** (**a**) Bouguer gravity map; (**b**) free-air gravity map; and (**c**) total intensity magnetic anomaly map of the area investigated.

## 4. Discussion

The marine gravity anomalies in the area under investigation tended to have a direct impact on the bathymetric changes adjacent to the coast. These included major Bouguer gravity and free-air gravity anomalies ranging from −40 to 200 mGal and −130 to 90 mGal, respectively (Figure 2a,b). The interpretation of these anomalies with regard to the main bathymetric features and geological tectonic setting countenance is presented herein. These essential gravity indicators referenced the Eratosthenes Seamount, Herodotus Basin, Levant Basin, Nile Delta cone and East Mediterranean Ridge, where the slope causes the bathymetric and topographic features to vary over short distances (Figure 1b). Furthermore, the regional–residual isolation technique was applied to analyse the potential field data in this study. The polynomial trend surface technique was applied to separate the residual from the regional fields provided by the Generic Mapping Tools (GMT) software [31]. Figure 3a,b shows the regional and residual gravity anomaly maps, respectively, which were created using a contour interval of 10 mGal for a clear display. Additionally, the regional–residual

magnetic maps are shown in Figure 3c,d, respectively. The qualitative dissection of the marine gravity and magnetic anomalies revealed the following features:

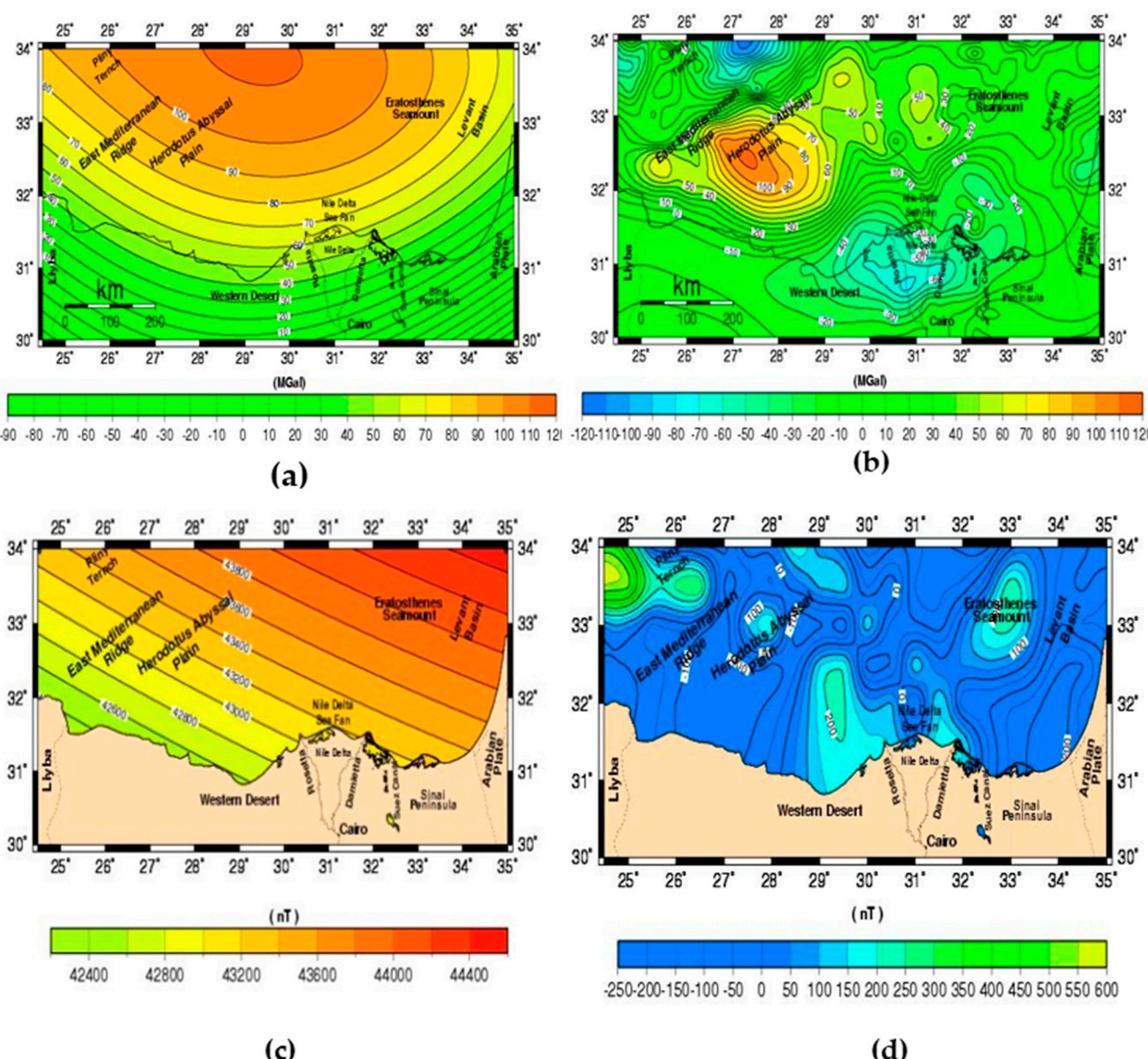

**Figure 3.** (**a**,**b**) Regional and residual gravity maps; (**c**,**d**) regional and residual magnetic maps of the area investigated.

The gravity map (Figure 2a) displays the layer density distribution directly concerning the deposit sequence of the area under investigation. A general feature of the map is the major gravity anomalies spread inside, with the exclusion of the southern portion. A positive gravity value (200 mGal) was eminent at the northwestern portion of the study area, but a negative (−Ve) gravity value (−40 mGal) was found at the southern portion.

Positive Bouguer anomalies predominated in the extended northeast–southwest direction together through a high of almost +200 mGal extending along the Herodotus abyssal plain. Lengthy anomalies were obvious and could be clearly identified as tight and raised gradient anomalies. The wide variation in the Bouguer gravity might be due to variation in the crust thickness, a property that concerns the Nile deep sea fan.

The gravity anomaly map revealed additional marine features associated with geological structures at the Levant Basin, Nile deep sea fan, Herodotus Basin and Eratosthenes

Seamount (Figure 2a). The study area was characterized by −Ve anomalies in the northeastern region of the Nile deep sea fan. Moreover, we observed a significant anomaly above the entrance of the Nile deep sea fan, which we concluded to be caused by the formation of deltaic deposits. Gravity anomaly maps show the isostatic stability of major scale structures found in geological basins (Figure 2b); the gravity anomaly map is high if the structure is only partially compensated for or not at all. The gravity anomaly map provides evidence of arranged tectonic structures at the regional and limited scale, thus revealing the influence of bathymetric and topographic features in an overall approach (Figure 1b).

A nearly constant concave −Ve anomaly arrangement was observed spreading lengthways from the northwest to the south at the middle portion of the study area (Figure 2b). The free-air anomalies and the bathymetric and topographic features exhibited numerous matches, i.e., the slopes of the free-air pattern were large patterns in the southeastern portion of the area (Figure 2b).

The total intensity of the magnetic anomalies ranged from +44,800 to +41,400 nT, as shown in Figure 2c. In general, closer and extended shape features were present in the magnetic anomalies. The magnetic anomalies tended to be oriented in the NW–SE direction, coinciding with the bathymetric and topographic features. Magnetic anomalies with high values were present in the northeastern part of the study area towards the Cyprus arc (Figure 2c), which were created by the ophiolites situated inside the sedimentary sequence. The low values of the magnetic anomalies at the Egyptian coast and under the Levant Basin were correlated with the thick sediments in these areas. The regional gravity anomaly values declined to the east–west and southeastern directions (Figure 3a). These reveal the consequence of the transition beginning from the oceanic to the continental crust in the study area towards the Arabian plate. The residual gravity anomalies reveal the outcome of the dissimilarity between the density of the sediments and the basement (Figure 3b). The regional magnetic anomalies showed major NW–SE trends, which increased in the northward direction; this may be attributed to the shallow depth of the basement rocks (Figure 3c). Figure 3d shows the residual magnetic anomalies. It is characterized by wide-ranging positive and −Ve anomalies covering large areas. The special belongings of thick sediment deposits and the remnant magnetization of the oceanic crust may cause a −Ve magnetic anomaly in the area offshore from Egypt.

## 5. 2D Modelling

A 2D gravity model was generated using the computer software program Talwani 2D gravity modelling (TWGRAV), as reported in [32]. The 2D gravity forward modelling of this study was performed according to the methods described in [32], which incorporates 2D formulations from [33]. For the 2D gravity modelling, the free-air gravity anomalies were applied where the Bouguer gravity data may have encompassed further inaccuracy. On behalf of its design, the Bouguer gravity field in deep water is considerable. Bathymetry laterally profiles the model regions, which are not sufficiently fitted, and the Bouguer gravity field may cover this incorrectness. Generally, several geological structures are somewhat stringy, and the difficulties associated with them can be explained with 2D modelling procedures. Several approaches intended for the calculation of the gravitational attraction instigated by occasionally formed 2D modelling bodies have been introduced [27,32,34–37]. These approaches use a 2D modelling system procedure based on a geological cross unit, wherein the forms of the effects of gravity are planned to shown by a polygon. The 2D modelling is split into numerous small bodies of dissimilar sizes, followed by similar forms [32]. In this technique, the 2D modelling is estimated using a polygon with a sufficiently large number of borders.

Together, the vertical–horizontal constituents of the gravitational attraction caused by this polygon can be calculated at some specified point. In addition, the densities of 2D modelling were controlled by the Vp velocities in [38] and Birch empirical functions in [39,40]. These studies established an experimental correlation among seismic velocities and densities. This correlation was initially created between shelf and deep-sea deposits;

however, it was stretched to a broader variety of rocks with greater velocities. In a Birch relation was applied to adapt the Vp velocity into the density (*p*). Table 1 summarizes the density values that were useful in the designs [38,39]. The 2D modelling body was designed lengthways with three seismic profiles (P1–P1″, P2–P2″, and P3–P3″). These profiles intersected the bathymetric–topographic features of the area under investigation (Figure 1b). These seismic outcomes can be used to make an early density model. Free-air anomaly standards were tested every half kilometre along the seismic profiles to achieve an even 2D (Figure 2b). The 2D modelling procedures carried out on the three profiles are presented in Figures 4–6.

**Table 1.** Density and velocity relationship in g/cm$^3$ unit and (km/s), respectively. Adapted from Birch empirical functions [38,39].

| Layer | Density (*p*) Model (g/cm$^3$) | Velocity Relationship (Vp) (km/sec) |
|---|---|---|
| Water | *p* = 1.04 | Vp = 1.4–1.5 |
| Sediments | *p* = 2.35–2.5 | Vp = 1.5–4.5 |
| Upper crust | *p* = 2.75–2.82 | Vp = 6.0 |
| Lower crust | *p* = 2.90 | Vp = 6.5 |
| Oceanic crust | *p* = 2.92–2.95 | Vp = 7.0 |
| Moho | *p* = 3.30 | Vp = 8.0 |

Figure 4 shows the 2D gravity modelling alongside the profiles P1–P1″ in portion (C). The model was about 285 km long. The bathymetric value of the profile is similarly displayed in Figure 4A for assessment. In addition, the free-air gravity anomalies designed from the 2D model and the detected anomalies are shown in Figure 4B. In the lateral profile, the gravity values exhibited significant dissimilarities, particularly near the left brink of the profile. A −Ve gravity anomaly was observed through the thickening of the deposited strata. The gravity values declined sharply near the Pliny trench area due to major crustal thickening and the marine geological features gradient. The basement deepness differed from approximately 9 km at the Egyptian coastline to around 13 km in the middle portion of the study area. The Moho depth next to the Egyptian coastline was approximately 23 km. The depth ranged between 20 and 24 km underneath the East Mediterranean Ridge and the Herodotus abyssal plain. These results agree with the seismic outcomes reported in [41]. The transition from the continental to oceanic crust inside the African plate was observed as an expanse around 40-km offshore from the Egyptian coast (Figure 4), agreeing with drilling evidence of sedimentary structures at the Egyptian coastline [3,6,9,14,42].

Figure 5C displays the 2D gravity modelling alongside profile P2-P2″. The profile was 240 km long, generally trending in the WNW–ESE direction and ranging from the Eratosthenes Seamount to the Levant Basin. The gravity anomalies designed from the 2D model and the observed field are shown in Figure 5B. The bathymetry along this profile is accessible as shown in Figure 5A. Moreover, at the profile's limits, the −Ve values ranged from −0.5 to −60 mGal and displayed a rapid reduction, as shown in the free-air anomaly alongside profile P2–P2″. This is indicative of a severe dewdrop of the basement and the growth of the sedimentary layer; in this case, there was an abundant deposit cover located underneath the Plio Quaternary sediment in the Levant Basin. The movement of the oceanic-continental crust happens at a distance of approximately 60 km from the Arabian plate, as determined by the seismic results (Figure 5). The depth of the basement extends from around 4 km underneath the Eratosthenes Seamount to around 13 km beneath the Levant Basin, and decreases to approximately 6 km near the Arabian plate. The Moho depth varies from approximately 26 km underneath the Eratosthenes Seamount to around 23 km below the Levant Basin, and to approximately 24 km towards the Arabian plate.

Figure 6C displays the 2D gravity modelling alongside profile P3–P3″. The bathymetric rate of the profile was too accessible in Figure 6A. The gravity anomaly calculated by the 2D model, which resulted from the seismic model, is presented in Figure 6B. The profile was 400 km long, and the free-air anomaly had −Ve values ranging from −2 mGal to −140 mGal below the East Mediterranean Ridge. The Moho depth values were significantly

associated with the dissimilarities in the gravity values alongside the profile. The Moho site had a depth of around 27 km near the Egyptian coastline, which increased as a slightest depth of around 19 km below the Herodotus abyssal plain. The Moho depth increased again around 26 km under the East Mediterranean Ridge. The rise in the −Ve gravity value resembles the comparatively great depth of the sedimentary deposit. The evolution of the continental and oceanic crust inside the African plate was modelled at an extant beyond around 40 km from the Egyptian coastline [43]. The basement depth of approximately 8 km under the Egyptian coastline agreed with the borehole outcomes presented by [42]. The depth ranged from about 10 to 13 km at the Herodotus abyssal plain and increased to 14.5 km below the East Mediterranean Ridge. The major part of the seismic section data in 2D gravity modelling along profiles P1–P1″, P2–P2″, and P3–P3″ as shown in (Figures 4–6) are controlled as described in [26,44,45].

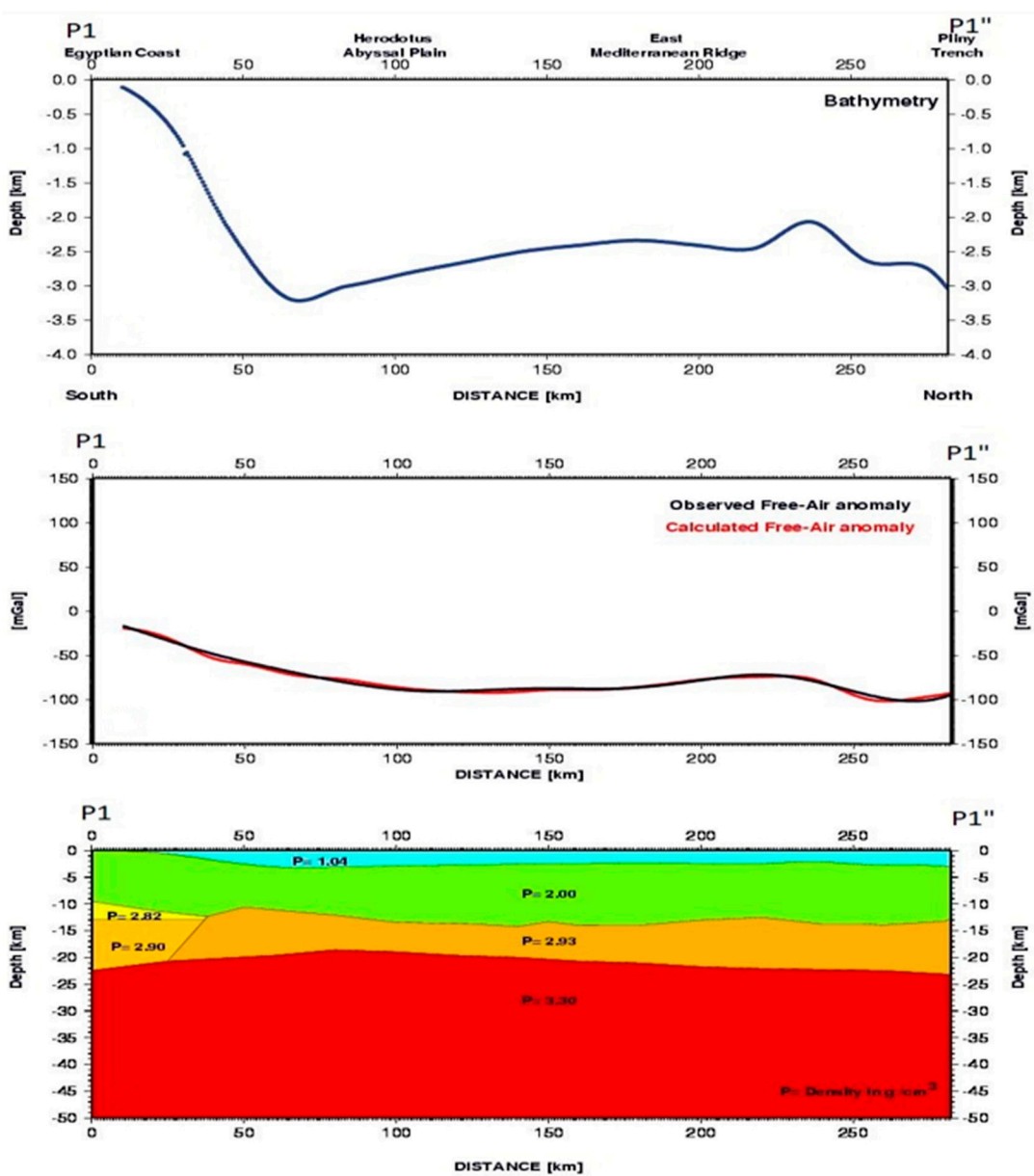

**Figure 4.** 2D modelling gravity along profile P1–P1″.

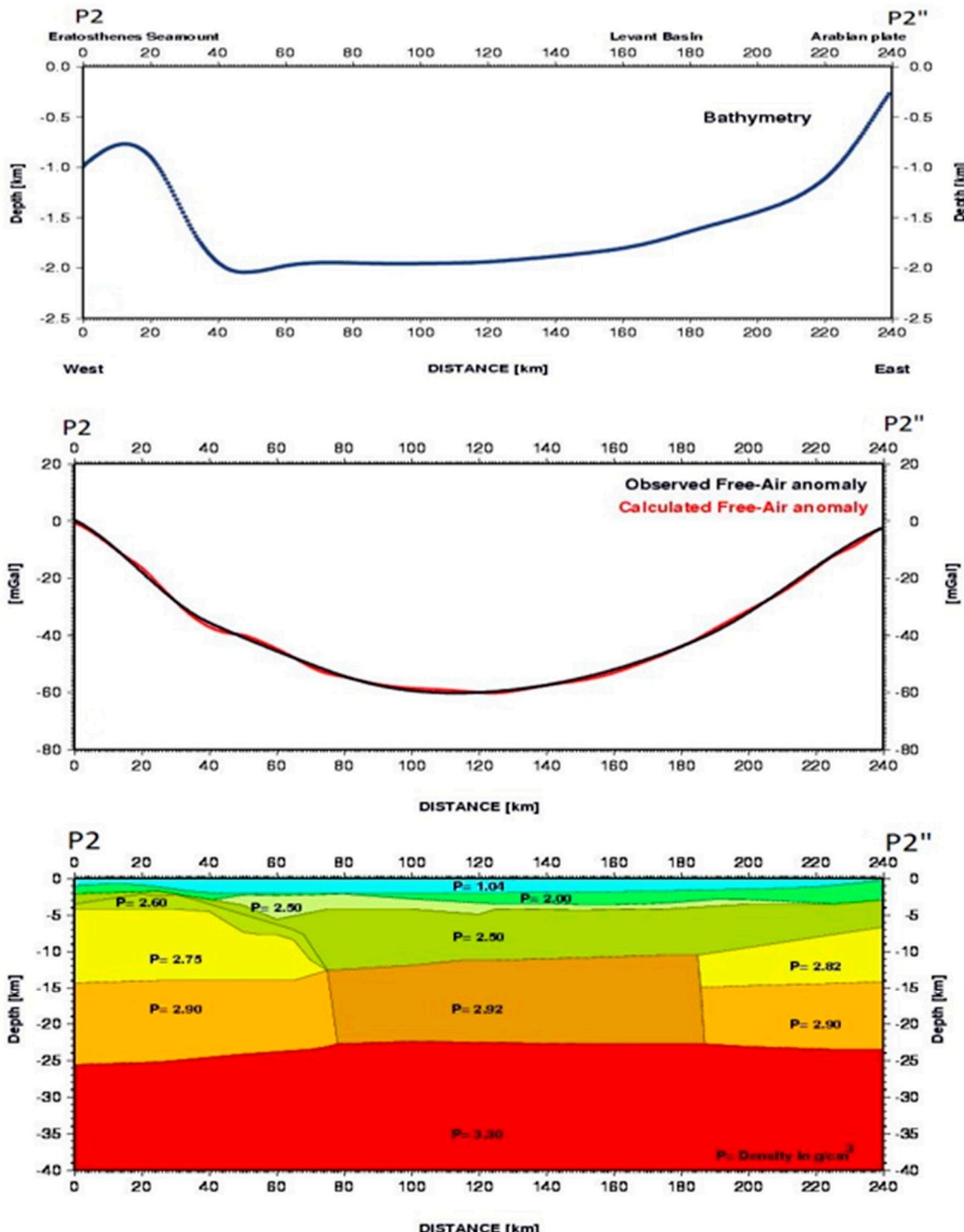

**Figure 5.** 2D modelling gravity along profile P2–P2″.

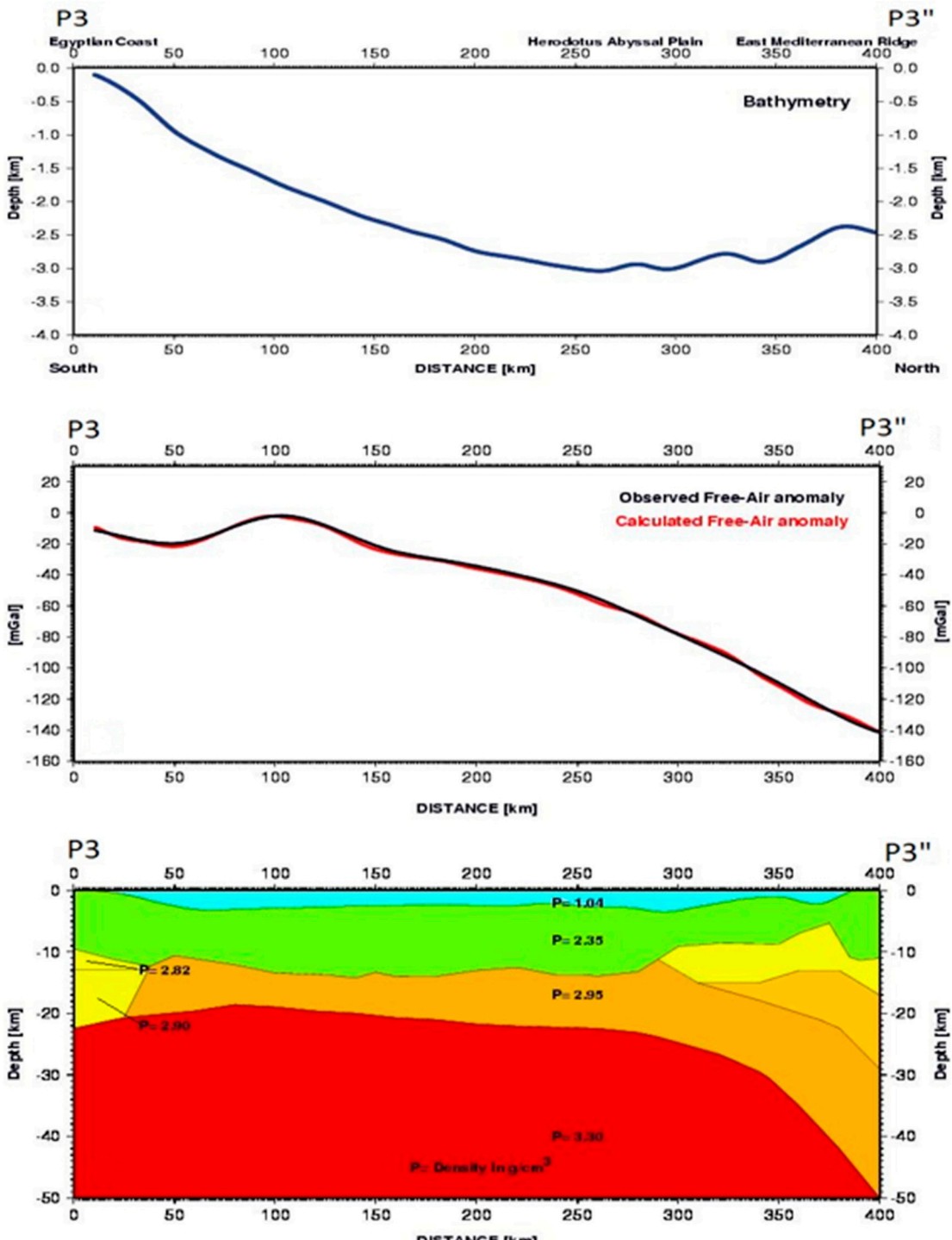

**Figure 6.** 2D modelling gravity along profile P3–P3″.

According to the main results achieved in the current study, a drawing model of the study area and its adjacent areas showing a simplified geological tectonic model of the major geological structure was produced and is presented in Figure 7a. A sketch model is additionally based on the results of previous various geological studies inside the study area [8,18–22]. For the overall study area as illustrated in Figure 7a–e, it seems that:

1.  The transition from the continental-oceanic crust inside the African plate was detected as an expanse around 40 km offshore from the Egyptian coast (Figure 7b).

2. The Eastern Mediterranean region includes a short segment of the subduction zone boundary between Africa and Eurasia (Figure 7a).

3. Figure 7a showed the (EMTS) runs the base of the continental margin of (E. Libya and W Egypt) through the head of the Nile Delta and in the long run into the Gulf of Suez as stated by [22]. Furthermore, the sketch simplified of the tectonic model of the active EMTS at the Egyptian continental margin is illustrated in Figure 7e.

4. In the southeast of the study area, the triple junction (Africa, Arabian, and Sinai) is sited as proposed by [20] (Figure 7a).

5. The intersection among the (EAFZ), and the (DSF) characterize a new triple junction zone along the EAFZ between Africa, Anatolian, and Arabian plates (Figure 7a), as proposed by [9].

6. 2D modeling results recognize the continental-oceanic crust transition at E. Mediterranean Ridge and Levant Basin (Figure 7c,d).

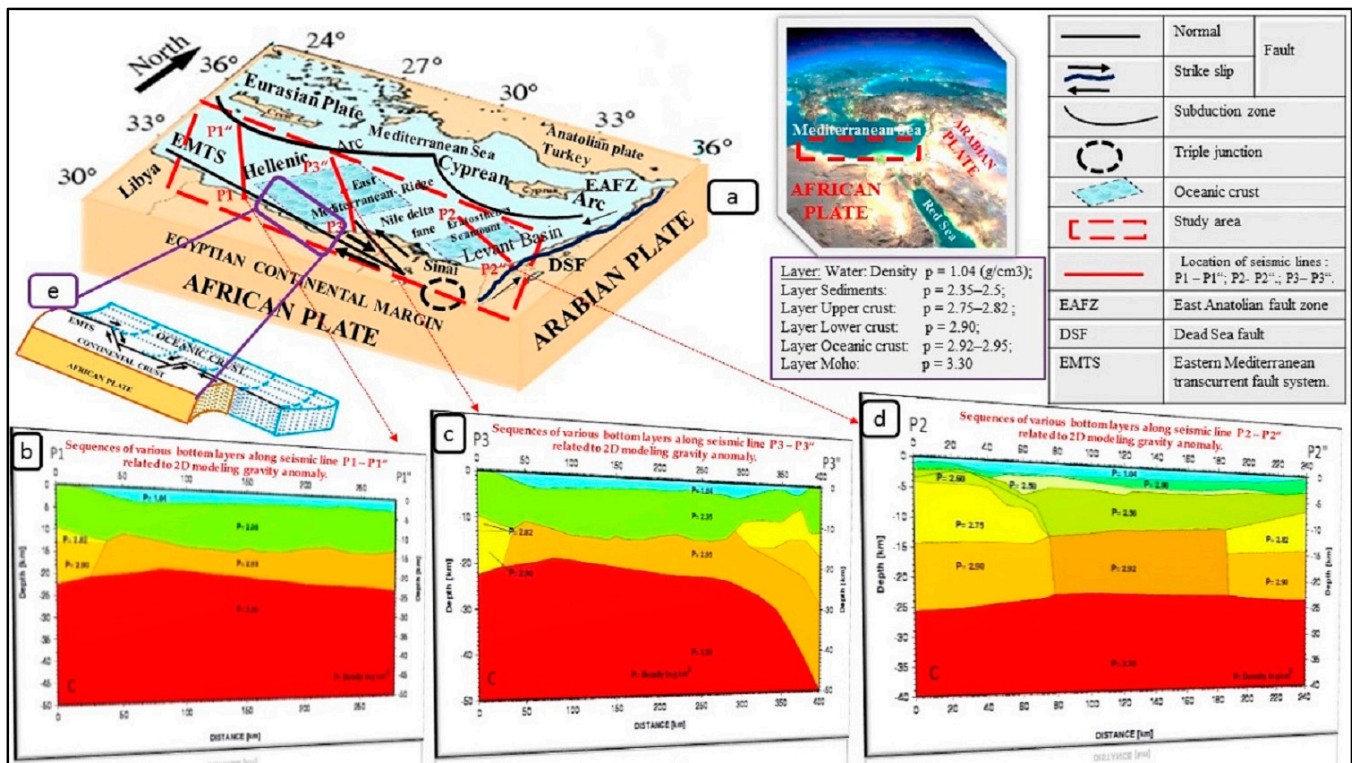

**Figure 7.** (**a**) Showing simplified geological tectonic model of the major geological structure in the study area and its adjacent areas. Based on [8,9,18–22]. (**b**–**d**) The main results of the current study showing sequences of various bottom layers along seismic lines P1–P1″, P2–P2″, and P3–P3″ related to 2D modeling gravity anomaly, respectively (See, Figures 4–6). (**e**) Showing a tectonic model of the active EMTS at the Egyptian continental margin.

## 6. Conclusions

(1) In general, the considered models revealed that the southeastern portion is oceanic crust, which is exposed by approximately 13-km thick sediments.

(2) The transition from continental to oceanic crust was also observed at approximately 56 km from the Arabian Plate, with the continental crust mainly found beneath the Eratosthenes Seamount.

(3) The gravity model was used to assess the density disseminations of the deep lithospheric structures along the northern Egyptian passive continental margin.

(4) The marine gravity data investigation revealed the bathymetry features of the oceanic crust and the intra crustal homogeneities within it. An examination of the gravity

data was carried out to develop the geometry and the density spreading in the 2D designed profiles.

(5) The results of this study provide a significant contribution to knowledge on the inferred landscape and proper crustal structure of the northern Egyptian passive continental margin.

(6) The 2D modelling of gravity was adapted so that the calculated gravity anomalies were adjacent to those detected without any substantial variations of the seismic achieved geometry. The synthesis of the analysis and results leads to the following conclusions:

- There is a transition from the continental to oceanic crust at the Levant Basin about 56 km from Arabian Plate and about 40 km from the Egyptian coast.
- The modelled and observed gravity field showed good agreement, including known regional structures.
- The model has shown evidence on the crustal structure caused by the severe tectonic activities and the great thickness of sedimentary layers, which cover the crystal line crust.
- The 2D gravity modelling results fit into the existing geodynamic and tectonic framework.
- Finally, more advanced geological and geophysical studies are important for present assembly projects to identify recent tectonic movements inside the area under investigation.

**Funding:** This research received no external funding.

**Institutional Review Board Statement:** Not applicable.

**Informed Consent Statement:** Not applicable.

**Data Availability Statement:** The study did not report any data.

**Acknowledgments:** The author wish to thank the database (B.G.I.), Avenu Edourad Belin, France, for giving the geophysical field data. It is my pleasure to thank EGPC, 1980, for providing a set of Bouguer gravity anomaly maps of Egypt at 1:500,000 scale. The authors also wish to thank and encompass their sincere appreciation to the Geological Survey of Cyprus for providing deep seismic sounding experiments along three profile seismic regional constructions in the study area. The figures have been created by the (GMT). I am very grateful to the editor and the reviewers for their constructive comments and editorial handling.

**Conflicts of Interest:** The author declares no conflict of interest.

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
