# Peer review of "Marine Geophysical Data and Its Application to Assessment of Crustal Structure along the Northern Egyptian Passive Continental Margin"

_applsci, doi:10.3390/app122311901_

Round 1

Reviewer 1 Report

Dear Editor-in-Chief and authors:

This is a Comment on “Marine geophysical data and its application to assessment of crustal structure along the northern Egyptian passive continental margin”.

This paper used accessible potential field data (marine gravity data) to provide a wider vision of the potential field of the area under investigation. Two-dimensional (2D) gravity modeling indicated the respectable agreement between the modeled gravity and observed gravity field, including known regional structures found in the investigated area, and also revealed that the crustal modeling was affected by the tectonic structure and the big thickness of sedimentary layers, which barrier the crystal crust. The authors also suggested that the basement extent lies nearly 6-9 kilometers lower in the northern Egyptian coastline to approximately 13 kilometers under the Herodotus abyssal plain. This is a nice and lovely paper, I suggested a minor revision is needed.

Comments: (1) I saw many previous scholars also studied this area (your cited paper 1 to 13), in the introduction part, the present of the previous controversy is not clear, I hope the authors keep more attention to a scientific problem or a technological problem in geophysical inversion. (2) English writing should be improved,  for example, in the abstract, you present, the result, the result, the result. The geophysical inversion method you used, you also have to present. You used two moreover, that is not necessary. (3) the pictures you presented in the last are not so clear, it does not match their caption, and the last picture, no caption? (4) I suggested that the author put the result into the “DATA PROCESSING METHODS AND its result”, the discussion part should be individually stated, and you need one or two specious arguments in your discussion part, and you can make your advantage of the geophysical inversion method. (5) short the final conclusion part, the first paragraph in your conclusion part is too long.

Cheers!!! 

Author Response

Response to Reviewer 1 Comments

Comments and Suggestions for Authors

Point (1): I saw many previous scholars also studied this area (your cited paper 1 to 13), in the introduction part, the present of the previous controversy is not clear, I hope the authors keep more attention to a scientific problem or a technological problem in geophysical inversion.

Response for point (1): I did follow the comments and this is explained as you recommended and adjusted as yours recommended in the manuscript  (in the Red Line) despite the conclusions provided in these studies, there are still uncertainties about the underlying crustal structure in the study area; Meanwhile, the character of the dissimilar tectonic units is still up for debate. In general, as I mentioned in the manuscript, various previous studies have shown that the area under investigation appears to be characterized by changeability in the crustal construction due to the subduction of the sea bottom. This motivated me to improve the results of previous studies on the study area. For that, the aim of this work was to achieve 2D gravity modeling data and define the level of crustal structure and its thickness. Currently, this is the first goal, and later in another publication, I hope will do something with 3D modeling data if the program is available in the study area. Moreover, I stated in a manuscript that the approaches to use a 2D modeling system procedure to do technological in geophysical inversion are planned to be shown by a polygon. (Talwani, et al, 1965).

Point (2): English writing should be improved, for example, in the abstract, you present, the result, the result, the result. The geophysical inversion method you used, you also have to present. You used two moreover, that is not necessary.

Response for point (2): I did your recommendation using a professional English editor to check the manuscript "Please see the attachment." English-Certificate of Editing-Job_149088

Point (3): The pictures you presented in the last are not so clear; it does not match their caption, and the last picture, no caption?

Response for point (3): This part is reformulated and adjusted as yours recommended in manuscript.

Point (4): I suggested that the author put the result into the “DATA PROCESSING METHODS AND its result”, the discussion part should be individually stated, you need one or two specious arguments in your discussion part, and you can make your advantage of the geophysical inversion method.

Response for point (4): This part is reformulated and adjusted as yours recommended in manuscript

Point (5): Short the conclusion part, the first paragraph in your conclusion part is too long.

Response for point (5): This part is reformulated and adjusted as yours recommended in manuscript.

Note: I did your recommendation using a professional English editor to check the manuscript "Please see the attachment." English-Certificate of Editing-Job_149088 - (in Upload File)

Reviewer 2 Report

The work is interesting. It is dedicated to the structure of the earth's crust of the Mediterranean Sea in the area of ​​the coast of Egypt. Such non-standard issues as the place of transition of the continental crust into the oceanic one and the thickness of sedimentary deposits are being solved. This is of great importance for the search for hydrocarbon deposits.

However, there are a number of remarks.

The article is not framed according to the rules - there are no authors and affiliations immediately after the title.

llustrations and tables are not placed in the appropriate parts of the manuscript, but are grouped at its end.

It is not clear whether the authors used any methods of geophysical research. They are not described in the article. Either the authors limited themselves to interpreting the borrowed data.

In addition to bathymetric and geophysical maps, I would like to see a map of the tectonics of the studied part of the Mediterranean Sea.

Author Response

Response to Reviewer 2 Comments

Comments and Suggestions for Authors

Point (1): The article is not framed according to the rules - there are no authors and affiliations immediately after the title.

Response for point (1): This is reformulated and adjusted as yours recommended in manuscript.

Point (2): Illustrations and tables are not placed in the appropriate parts of the manuscript, but are grouped at its end.

Response for point (2): This part is reformulated and adjusted as yours recommended in manuscript.

Point (3): It is not clear whether the authors used any methods of geophysical research. They are not described in the article. Either the authors limited themselves to interpreting the borrowed data.

Response for point (3): I did follow the comments and this is explained as you recommended in the manuscript. In the part of data and processing methods was state that, several accessible data sets, including marine gravity data, seismic data, bathymetric and topographic data, magnetic data, and other ancillary data, were sensibly revised and reviewed to achieve the aims of the current study.

Point (4): In addition to bathymetric and geophysical maps, I would like to see a map of the tectonics of the studied part of the Mediterranean Sea.

Response for point (4): I adjusted as yours recommended in manuscript as sketch showing regional tectonic setting in the study area and its environs, modified from, and the lower diagram shows a schematic cross-section along the line indicated in the upper diagram (Fig. 1 c and d) modified from [15].

Note: I did your recommendation using a professional English editor to check the manuscript "Please see the attachment." English-Certificate of Editing-Job_149088 - (in Upload File)

Round 2

Reviewer 2 Report

There is no model of the geological structure of the bottom of the coastal part of the Mediterranean Sea either in text or in graphical form

Author Response

Review Report (Round 2)

Response to Reviewer 2 Comments

Comments and Suggestions for Authors

Point (1):  There is no model of the geological structure of the bottom of the coastal part of the Mediterranean Sea either in text or in graphical form

Response for point (1):

  • Firstly, I really extend my thanks and appreciation to you for your valuable comments in general.
  • I reformulated and adjusted as you recommended to create a model of the geological structure of the bottom of the coastal part of the Mediterranean Sea. You find it in the manuscript as text (Red Line) and illustrated in figure 7 (a, b, c, d, e) as the following:

According to the main results achieved in the current study, a drawing model of the study area and its adjacent areas showing a simplified geological tectonic model of the major geological structure was produced and is presented in figure (7a). A sketch model is additionally based on the results of previous various geological studies inside the study area [8, 15–19].  For the overall study area as illustrated in figure 7.1, it seems that:

  • The transition from the continental-oceanic crust inside the African plate was detected as an expanse around 40 km offshore from the Egyptian coast (Fig. 7b).
  • The Eastern Mediterranean region includes a short segment of the subduction zone boundary between Africa and Eurasia (Fig. 7a).
  • Figure (7a) showed the (EMTS) runs the base of the continental margin of (E. Libya and W Egypt) through the head of the Nile Delta and in the long run into the Gulf of Suez as stated by [19]. Furthermore, the sketch simplified of the tectonic model of the active EMTS at the Egyptian continental margin is illustrated in figure (7e).
  • In the southeast of the study area, the triple junction (Africa, Arabian, and Sinai) is sited as proposed by [17] (fig. 7a).
  • The intersection among the (EAFZ), and the (DSF) characterize a new triple junction zone along the EAFZ between Africa, Anatolian and Arabian plates (Figure 7a) as proposed by [9].
  • 2 D modeling results recognize the continental-oceanic crust transition at E. Mediterranean Ridge and Levant Basin (Fig.7c, d).
